# Astringency Sensitivity to Tannic Acid: Effect of Ageing and Saliva

**DOI:** 10.3390/molecules27051617

**Published:** 2022-02-28

**Authors:** Mei Wang, Chantal Septier, Hélène Brignot, Christophe Martin, Francis Canon, Gilles Feron

**Affiliations:** Centre des Sciences du Goût et de l’Alimentation, AgroSup Dijon, CNRS, INRAE, Université Bourgogne Franche-Comté, F-21000 Dijon, France; mei.wang@inrae.fr (M.W.); chantal.septier@inrae.fr (C.S.); helene.brignot@inrae.fr (H.B.); christophe.martin@inrae.fr (C.M.); francis.canon@inrae.fr (F.C.)

**Keywords:** astringency, threshold, saliva, elderly, proline-rich-protein, tannic acid

## Abstract

Astringency is an important sensory characteristic of food and beverages containing polyphenols. However, astringency perception in elderly people has not been previously documented. The aim of the present work was to evaluate sensitivity to astringency as a function of age, salivary flow and protein amount. Fifty-four panellists, including 30 elderly people (age = 75 ± 4.2 years) and 24 young people (age = 29.4 ± 3.8 years), participated in this study. Astringency sensitivity was evaluated by the 2-alternative forced choice (2-AFC) procedure using tannic acid solutions. Whole saliva was collected for 5 min before and after the sensory tests. The results showed that the astringency threshold was significantly higher in the elderly group than the young group. No correlation was observed between the salivary protein amount and threshold value. However, a negative correlation between salivary flow and threshold was observed in the young group only. These results showed a difference in oral astringency perception as a function of age. This difference can be linked to salivary properties that differ as a function of age.

## 1. Introduction

Dietary polyphenols are a class of compounds present in foods and beverages, such as vegetables, nuts, unripe fruits and berries, wine, tea, etc. [1,2,3], and they are of great interest for the food industry because of their potential beneficial effects on health, particularly for the ageing population [4,5,6]. In food and beverages, polyphenols, especially tannins, can elicit astringency, which is perceived as a quality parameter and desired at balanced levels depending on the food products [7,8,9,10]. In contrast, above a certain intensity, astringency is usually described as an unpleasant oral sensation [11,12], which limits the use and promotion of polyphenols at moderate levels in food despite their health benefits [9,13].

In 2004, the American Society for Testing and Materials (ASTM) defined astringency as “the complex of sensations due to shrinking, drawing or puckering of the epithelium as a result of exposure to substances such as alums or tannins” ([14] cited by [12]). Astringency is not confined to a particular region of the mouth but is a diffuse sensation [15]. Astringency is generally considered to be a tactile sensation detected through the activation of mechanoreceptors rather than a taste [16]. Indeed, astringency takes 20 to 30 s to develop fully, often being the last oral sensation detected [15]. Although the mechanism of astringency is not yet well understood, several hypotheses have been proposed in the literature to explain astringency onset [12,16,17]. It is most probably detected at the level of the oral mucosa [9], either by mechanoreceptors after the increase of the friction forces at the surface of the epithelial cells [18] or by the detection of the aggregation of the mucosal pellicle by the transmembrane mucin MUC1 as recently proposed by Canon et al. [16]. Salivary proteins are thought to play an important role in these two hypotheses by protecting the mucosal pellicle from aggregation by tannins. Indeed, their presence, and especially those of tannin-binding proteins, such as proline-rich proteins, decrease the perception of astringency [19,20].

Regarding the effect of ageing on astringency perception, the literature is quite scarce, although the influence of ageing on the perception of other taste modalities has been largely documented. Longitudinal and cross-sectional studies have found that taste and smell losses tend to become noticeable after 60 years of age, with greater severity after 70 years of age [21,22]. In 2012, a systematic review and meta-analysis showed that most of the primary studies included in the review (n = 69) observed an increase in taste detection and identification thresholds and a decrease in taste intensity at the supra-threshold levels for the five basic taste modalities (bitter, salt, sour, sweet, umami) [23]. However, the authors highlighted the lack of concordance among the primary studies regarding the extent of taste loss. This discrepancy between studies is probably due to significant differences in the sensory procedures used to evaluate taste acuity [23,24]. More recently, Doty et al. (2018) [25] evaluated a decline in the five basic taste perceptions in 1020 Caucasian European subjects (age 18–80 y/o). The study confirmed taste losses with ageing regardless of the modality. The authors also highlighted the complexity of the association between the ability to perceive a taste and the preference for the same. Moreover, beyond this overall effect of age on taste abilities, ageing is also accompanied by large interindividual variability in olfactory performance scores and, to a lesser degree, in taste performance scores [26].

Several factors can influence the extent of sensory decline during ageing (nutritional status, general health and diseases) [27]. The reasons for these sensory modifications can also be linked to changes in oral physiology with age. Indeed, in the elderly population, the cumulative effects of physiological ageing, diseases and drugs frequently impact the different aspects of oral physiology that are of great importance in taste and aroma sensitivity and thus eating behaviour [27,28,29]. In particular, ageing may often be accompanied by a decrease in salivary flow or changes in salivary composition [30], which can lead to a dry mouth or xerostomia. Hyposalivation is common among older adults due to an age-related decline in salivary gland function, and other causes include medications and systemic diseases [31]. Recently, Descamps et al. [32] found an average 38.5% reduction in resting salivary flow and a 38% reduction in stimulated salivary flow in healthy elderly people compared to young adults. This salivary hypofunction in elderly individuals can lead to changes in aroma, taste and textural perception, and consequently, food intake and consumption [29,30,33,34,35].

In the context of the world population becoming older and ageing well, the main objective of this study was to investigate the sensitivity to astringency as a function of age and salivary characteristics (flow and protein amount). For this purpose, a 2-alternative forced choice (2-AFC) methodology was applied to estimate astringency sensitivity in young and elderly panels while evaluating salivary flow and protein amount. Relationships between salivary flow, protein amount and sensitivity to astringency as a function of age are discussed.

## 2. Results

### 2.1. Astringency Threshold

No significant differences were observed between the three sessions regarding astringency thresholds for either Group Y (young panel) (Friedman chi2 = 1.13, *p* = 0.56) or Group O (elderly panel) (Friedman chi2 = 1.14, *p* = 0.56). Therefore, we decided to merge threshold values into a unique variable.

A significant difference was observed between the Y and O groups (Z = −2.5, *p* = 0.0110). The O group showed a higher mean astringency threshold than the Y group (Table 1, Figure 1).

### 2.2. Salivary Flow Rate and Protein Amount

No significant differences were observed between sessions regarding SFStart and SFEnd for Group Y (SFStart: Friedman chi2 = 0.75, *p* = 0.68; SFEnd: Friedman chi2 = 0.75, *p* = 0.68) or Group O (SFStart: Friedman chi2 = 5.2, *p* = 0.07; SFEnd: Friedman chi2 = 1.3, *p* = 0.53) or between the mean SFStart and mean SFEnd for Group Y (Friedman chi2 = 0.68, *p* = 0.492) or Group O (Friedman chi2 = 1.49, *p* = 0.135). For this reason, we decided to merge both variables into a unique variable, i.e., mean salivary flow (SF). SF values are presented in Figure 2. With regard to the comparison of salivary flow rate, the SF in the O group was lower than that in the Y group but with a modest degree of evidence (Z =1.66, *p* = 0.09) (Table 1). Moreover, a larger variability was observed in the O group compared to the Y group, with the presence of outliers with a higher SF. This large interindividual variability was previously observed in a large panel of elderly subjects and can be explained by life-style and aging factors such as diet, smoking habits, hydration status or structural changes in the salivary glands [32].

No significant differences were observed between sessions regarding protein amount for Group Y (Friedman chi2 = 1.08, *p* = 0.58) or Group O (Friedman chi2 = 2.55, *p* = 0.28) or between the beginning and the end of the session for Group Y (Friedman chi2 = 1.5, *p* = 0.91) or Group O (Friedman chi2 = 1.70, *p* = 0.19). For this reason, we decided to merge the protein amount into a unique variable (Table 1). Protein amounts are presented in Figure 3, and no significant differences were observed between the Y and O groups (Z = −0.32, *p* = 0.74), which confirms previous results [36].

### 2.3. Correlation between the Astringency Threshold and the Flow Rate and Protein Amount

The Spearman correlation between threshold and SF was not significant in the whole panel or the O group (Table 2). However, a significant and negative correlation was observed in the young (Y) group (r = −0.44, *p* = 0.03), where a higher salivary flow was associated with a lower threshold (Figure 4).

The Spearman correlation between the threshold and protein amount was not significant in the whole panel, the Y group or the O group (Table 2).

## 3. Discussion

In the current study, the astringency threshold was higher in elderly participants than in young participants. In other words, young adults were more sensitive to astringency than elderly adults, which confirms the findings of previous studies for other taste modalities [23,24,25,27,37]. To the best of our knowledge, this is the first study evaluating sensitivity to astringency as a function of age. In 2017, Linne and Simmons [10] investigated the impact of age on individual sensitivities to lingual tactile roughness in relation to sensitivity to astringent stimuli. The authors did not find a correlation between age and roughness sensitivity. However, their group was younger (21 to 60 y/o, n = 30) compared with that in our study as well as most other studies reporting taste differences as a function of age.

We found average detection thresholds of 0.2 g/L and 0.41 g/L for the Y and O groups, respectively. Using similar sensory procedures and stimuli, Linne et al., 2017 [10], obtained a detection threshold of 0.212 mM (0.36 g/L), which is close to our results. The increase in the detection threshold between Y and O was 1.6-fold. Similar increases on average were described for other taste modalities, such as saltiness (1.5), sourness (1.5), sweetness (1.4), umami (2.2) and bitterness (1.2 to 4.1) [23], suggesting that astringency sensitivity loss with age is not unusual compared to these modalities.

Differences in salivary properties can explain differences in taste sensitivity. Indeed, saliva allows the transport of taste substances to the taste receptor and protects the receptors by providing growth factors for the renewal of taste buds [38,39]. Some salivary components can modulate taste sensitivity [40]. For instance, sodium and amino acid salivary concentrations can modulate the detection threshold. Salivary flow can also influence fat intensity perception and preference, and larger amounts of saliva contribute to a higher in-mouth washing of lipid emulsion when tasted [41,42].

Regarding astringency, modulation of its perception as a function of the salivary flow rate led to contradictory results. Indeed, it has been reported that subjects with low salivary flow rated astringency higher than subjects with high salivary flow [43]. Conversely, Fisher et al. (1994) and Smith and Noble (1996) did not observe a difference in intensity rating as a function of salivary flow using temporal perception experiments [44,45]. Finally, Linne et al. (2017) [10] reported a higher sensory threshold for tannic acid in subjects with low salivary flow than in subjects with high flow.

In the present study, a positive relationship between astringency sensitivity and salivary flow was observed in the young panel only, i.e., a higher salivary flow corresponds to a higher sensitivity, which is in accordance with the results of Linne et al. (2017) [10]. This relationship of flow rate to astringency sensitivity, as shown in Figure 4, might suggest protection through an interaction mechanism with salivary proteins rather than a simple dilution effect, as suggested in other studies [46]. However, the amount of proteins measured in saliva from the Y group was not correlated with astringency sensitivity, which is consistent with previous studies that did not observe relationships between salivary total protein content and intensity or time-intensity evaluation of astringency [43,47]. However, strong positive correlations of astringency time-intensity parameters with some salivary protein fractions suggested that differences regarding astringency sensitivity and salivary properties were linked to salivary protein composition rather than global protein amount [15]. Indeed, histatins, mucins and salivary basic proline-rich proteins (bPRPs) have been identified as potential contributors to astringency perception in humans, while their role in the underlying mechanism of this perception is still under debate [17,48]. In particular, PRPs and histatins are described as tannin binding-proteins with a high affinity for tannins [49,50]. PRPs are secreted by the parotid glands, bind and scavenge tannins [51], giving them the ability to protect the mucosal pellicle against tannin aggregation [18]. Thus, bPRPs are proteins thought to play a role in astringency perception in humans [12,18]. In rodents, their role is much clearer as their presence in saliva increases the linking of astringent solution [19]. Indeed, the secretion of PRP in the saliva of rodents is not constitutive and is induced by the diet. Moreover, rodents do not secrete histatins [52] and thus PRPs are the main tannin-binding proteins in their saliva.

We did not observe such a relationship between salivary flow and astringency sensitivity in the elderly group or a difference in protein amount between the Y and O groups, which should explain the difference in sensitivity between the two groups. However, lower salivary flow was observed in the O group than in the Y group, which is in accordance with previous studies [32,53] and should partly explain the sensitivity differences between the two groups. Our observations suggest that the role of saliva in astringency sensitivity as a function of age should also be linked to salivary composition and, in particular, peptides and proteins. Studies on changes in salivary composition in healthy elderly individuals are relatively scarce and present poor consistency among results [30]. Moreover, the direction of change (increase or decrease) depends on the proteins. For instance, amylase, lysozyme and IgA increase with age, while lactoferrin, glutathione, peroxidase activity and mucin levels decrease, with a large consensus for the latter [30]. Similarly, histatin levels were also observed to decline with age [54], which is an interesting finding based on the possible involvement of mucins and histatins in astringency sensitivity [12,16]. We suggest that a lower level of these classes of proteins in the saliva of the elderly population should impact astringency perception. With regard to PRP and bPRP, there is a paucity of information describing their salivary amounts during the human lifespan. Exploring salivary exocrine protein secretion in 220 adults, Baum et al. [55] did not find a change in PRP secretion during ageing, although this study considered only acidic PRP.

In conclusion, we found that the astringency threshold was higher in the elderly group than in the young group, and our results suggest that salivary properties differently influence astringency sensitivity as a function of age. A deeper characterisation of salivary composition, particularly regarding PRP, mucin and histatin levels in both populations, should be performed.

## 4. Limitations

This study presents some limitations.

Although a preliminary session was provided to the training subjects on the astringency modality, bias due to other attributes, such as bitterness or olfactory cues, cannot be ignored. Regarding the latter, a nose clip was not worn during the sensory test to avoid excessive fatigue for elderly participants. The same concern guided the choice to limit the number of tannic acid concentrations to 4 to avoid excessive presentation of samples to this population.

The number of subjects was defined to be at least 23 participants in each group following a power test based on the results from a preliminary study. It is likely that this number of subjects will not permit us to capture all the variability commonly observed in the elderly population. Thus, an astringency evaluation in a larger population should be performed before generalising our findings. Moreover, we observed in both groups (Y and O) a large variability regarding the salivary flow with the presence of outliers in the O group which led to non-normally distributed data. We cannot rule out the potential effect of these outliers on the statistical results despite the use of non-parametric statistics more robust to their presence [56,57]. However, it is important to note that sensory evaluation and salivary sampling were repeated 3 times in 3 different sessions during a short period, which ensured the good reliability of our results.

Some of the participants in the elderly group took drugs despite the low mean number (2) compared to what was commonly observed in this population (ranging from 2.9 to 3.7 medications [58]). Drug intake is known to cause sensory impairment, particularly in the aged population [22,58]. In our study, we chose subjects who were significantly older (mean age 75 y/o) than those in the literature and thus more likely to take medication, which increased the difficulty of limiting the inclusion of drug-taking participants.

Finally, we principally included participants with good oral health based on dental observations. However, we did not check for oral microbiota impairment, a factor that is commonly observed in the elderly population and that should also affect taste sensitivity [59].

## 5. Materials and Methods

This study was approved on 31 October 2019 by the Ethical Committee CCP Ile de France IV under the number 2019-A02434-53.

### 5.1. Materials

Solutions for rinsing consisted of 0.1% pectin (Sigma–Aldrich, Saint-Quentin-Fallavier, France) and 1% bicarbonate (Gilbert, France) dissolved in Evian water at room temperature.

Solutions for the sensory training session consisted of six taste solutions (salty, sour, sweet, bitter, umami, and astringent), and their compositions are detailed in Appendix A. Each solution was coded with random three-digit codes.

Solutions for astringency sensitivity evaluation consisted of four solutions with increasing tannic acid (Sigma–Aldrich, Saint-Quentin-Fallavier, France) concentrations (in g/L) with a multiple of 3.05, i.e., 0.02, 0.062, 0.188, and 0.574. These concentrations were chosen on the basis of preliminary experiments performed with a small internal panel of subjects (see Section 2.2 below). All samples were prepared in Evian water at room temperature 1 h before testing. Since potassium alum has not been allowed in sensory studies, tannic acid was used as a component to evaluate astringency because it has been described as less bitter than other polyphenols, such as gallic acid and catechin [60], and thus limits the confusion between astringency and bitter taste. This was confirmed during preliminary tests.

### 5.2. Sensory Analysis

Fifty-four panellists, including 30 elderly (O) people (age ≥ 65 y/o) and 24 young (Y) people (age ≤ 35 y/o), were recruited to participate in the sensory sessions. The panel is described in Table 1. The number of subjects that needed to be included to find a difference between the two groups regarding astringency perception was determined by a power test (power = 0.9, significance level = 0.05, alternative = “two-sided”). The power test was based on preliminary results obtained on an internal panel (mean threshold = 0.19 ± 0.17 g/L of tannic acid, n = 9). At least 23 subjects per group (Y or O) were necessary to observe a difference equal to one standard deviation between the groups. More subjects were recruited in case of defection, particularly for the O group. These size groups are in line with previous studies aiming at evaluating the effect of ageing on taste perception regardless of the modalities [23,24]. Elderly and young subjects had good oral health, with a number of functional posterior units above 7 [32]. Moreover, elderly subjects were autonomous persons living at home, had no cognitive disorders (Mini Mental State Examination (MMSE > 25 [61])), did not have complete or half-complete dental appliances and took an average of 2 drugs per day (median = 1).

### 5.3. Preliminary Session

The objective of this session was to be sure that subjects were able to (i) clearly identify and differentiate astringency from other sensory sensations, in particular sourness, bitterness and olfactory cues, and (ii) perfectly understand the procedure of the sensory test, i.e., the 2-AFC to be used later.

The session was divided into two parts. During the first part, subjects received 20 mL of tasting sample in a fixed order at room temperature in plastic cups coded with random numbers. They were instructed to put the samples into their mouths, swirl the sample gently in the mouth for 30 s, spit it out and judge which taste it was. Between samples, subjects rinsed their mouth with Evian water and then waited for 1 min before the next sample. The tasting sensations were saltiness, sweetness, sourness, bitterness and umami. Additionally, panellists were presented with tannic acid solution as an example of astringency.

In the second part, subjects were trained and familiarised for the 2-AFC procedure as described below.

During both parts of the preliminary session, there was a discussion between subjects and experimenters after each test. At the end of the session, all the panellists indicated that they were able (i) to clearly identify astringency from other sensory sensations and (ii) to perform the 2-AFC test properly.

### 5.4. Testing Session

All sessions were performed for 3 months between the middle of November and the end of January and between 2 and 6 p.m. to minimise seasonal and circadian rhythms as much as possible. Moreover, panellists were asked to not drink, eat or smoke 1 h before the session.

The whole session was conducted under red light at room temperature in a sensory room equipped with individual boxes.

At the beginning of each session, panellists were asked to taste a model tannic acid solution of 1.76 g/L so that they could identify astringency. Then, they rinsed their mouths with pectin, bicarbonate and Evian water and waited for a 3 min break before threshold evaluation. The objective of this rinsing procedure is to perfectly clean the mouth to have the most similar oral conditions when starting each test, and thus minimise carry-over effects between sample evaluations. Sodium bicarbonate recovers pH homeostasis, and pectin removes tannic acid from the oral mucosa due to its capacity to form complexes with polyphenols [12]. This rinsing procedure was found to be efficient in wine studies for in-mouth aroma release experiments [62,63] and, more recently, for the time sensory evaluation of astringency and aroma [64]. This procedure was chosen instead of other procedures, such as the milk rinsing procedure [65], because of the necessity to avoid any contamination of saliva samples by food proteins.

The astringency threshold was evaluated by a 2-AFC procedure with ascending concentrations of tannic acid. In each 2-AFC presentation, two samples were presented: a target sample and a control sample. Each 2-AFC test was performed 3 times, and the evaluation was performed 3 times in 3 different sessions. Paired samples (5 mL) were presented in balanced order following a Latin square design (Williams design) at room temperature in a white plastic cup coded with the letter A or B. The testing procedure started from the lowest concentration. Panellists were given the reference or stimulus sample. They were asked to put the samples into their mouth, swirl them gently around the mouth for 30 s and then spit them out. They rinsed their mouths with pectin and waited for 1 min before evaluating the second sample. After 30 additional seconds, the panellists were asked to indicate which sample was perceived as astringent. Then, the panellists rinsed their mouth as described previously.

The sensitivity level was reached when three correct answers from the same concentration were achieved. The best estimate threshold for each subject was evaluated as the geometric mean of the three correctly answered concentrations and the previous lower concentration. When the subjects correctly identified the lowest concentration (0.02 g/L), the geometric means were calculated between this concentration and the theoretical concentration below, i.e., 0.02/3.05 = 0.0065 g/L. In contrast, when subjects did not correctly identify the highest concentration (0.574 g/L), the geometric mean was calculated between this concentration and the theoretical concentration above, i.e., 0.574 × 3.05 = 1.75 g/L.

### 5.5. Saliva Collection

Whole saliva was collected after the panellists had rinsed their mouths with 0.1% pectin, 1% bicarbonate and water at the start (SFStart) and at the end (SFEnd) of the session. Saliva was collected by expectorating into a preweighed tube with a cap for 5 min as described previously [41]. After collection, the tubes were weighed and then stored at −80 °C. Flow rates were determined gravimetrically and expressed as grams per minute (g/min).

### 5.6. Protein Amount

Saliva samples were centrifuged at g for 15 min at 4 °C before analysis. The protein concentration was determined in the supernatant using the Bradford protein assay, with bovine serum albumin (BSA) used as the standard for calibration [41].

### 5.7. Statistical Analysis

Data showed the presence of outliers for all the variables. Moreover, normality assumptions were not met for the raw data and residues. We decided to keep all the data and to not violate normality assumption. We thus performed nonparametric analyses because they are adapted to non-normally distributed data and are more robust to the presence of outliers [56,57,66,67,68]. Mann–Whitney U tests were performed to evaluate differences between the Y and O subjects regarding sensory and salivary parameters. Wilcoxon tests were performed on the salivary parameters (flow and protein amounts) to evaluate differences between the start and the end of each session. Friedman ANOVA was conducted on the threshold and salivary parameter measurements to evaluate the differences between the three sessions. Spearman rank order correlations were performed for the whole group and in each group (Y and O) to evaluate the relationships between salivary and sensory parameters. The significancy was set at *p* < 0.05. These tests were performed using Statistica^®^ version 13.5.0.17 (1984-2018 TIBCO Software Inc, Palo Alto, California, USA).

## Figures and Tables

**Figure 1 molecules-27-01617-f001:**
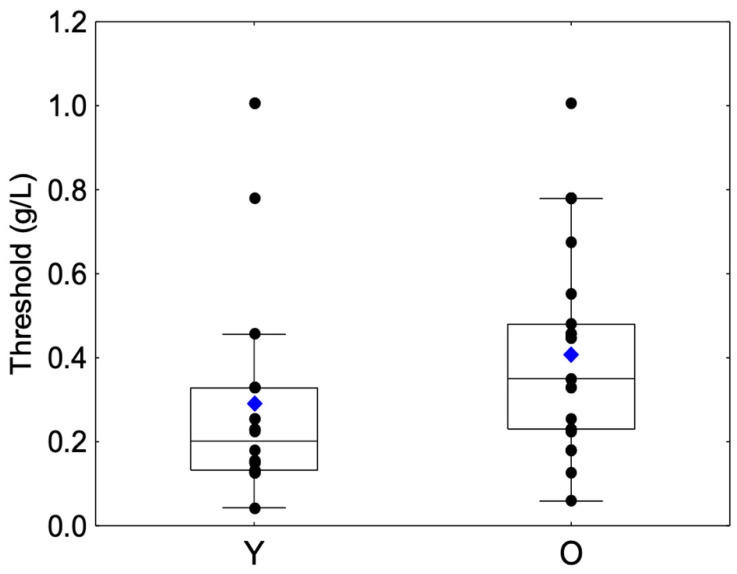
Box-plot distributions of threshold values as a function of the age category (young (Y) and elderly (O)) are shown. The bottom and top of the box correspond to the 25th and 75th percentiles, respectively. The horizontal band and the blue diamond correspond to the median and the mean, respectively. The ends of the whiskers represent nonoutlier ranges. The black dot symbols correspond to individual data points.

**Figure 2 molecules-27-01617-f002:**
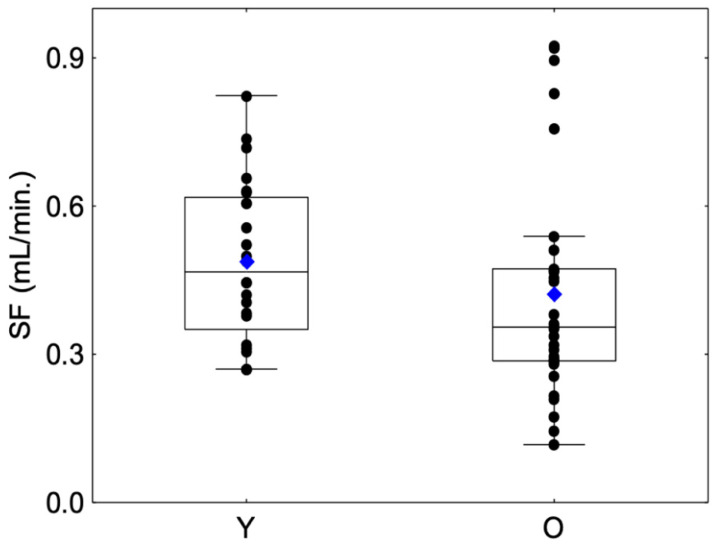
Box-plot distributions of whole salivary flow (SF) as a function of the age category (young (Y) and elderly (O)) are shown. The bottom and top of the box correspond to the 25th and 75th percentiles, respectively. The horizontal band and the blue diamond correspond to the median and the mean, respectively. The ends of the whiskers represent the nonoutlier range. The black dot symbols correspond to individual data points.

**Figure 3 molecules-27-01617-f003:**
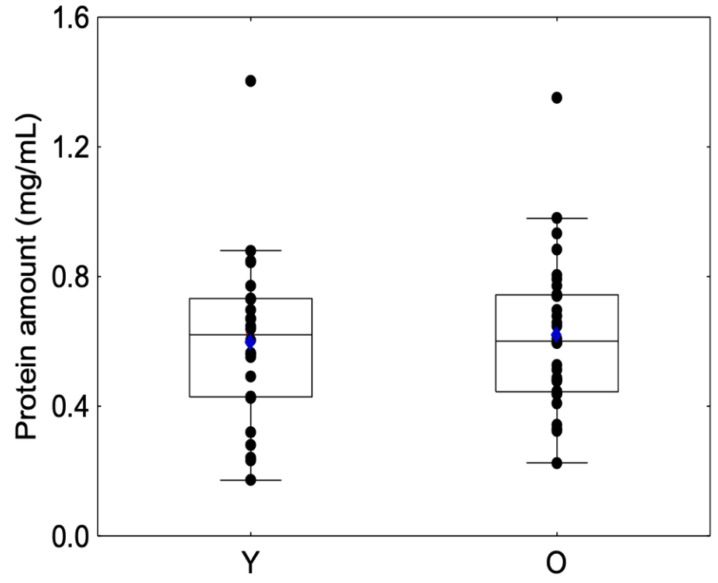
Box-plot distributions of salivary protein amount as a function of the age category (young (Y) and elderly (O)) are shown. The bottom and top of the box correspond to the 25th and 75th percentiles, respectively. The horizontal band and the blue diamond correspond to the median and the mean, respectively. The ends of the whiskers represent the nonoutlier range. The black dot symbols correspond to individual data points.

**Figure 4 molecules-27-01617-f004:**
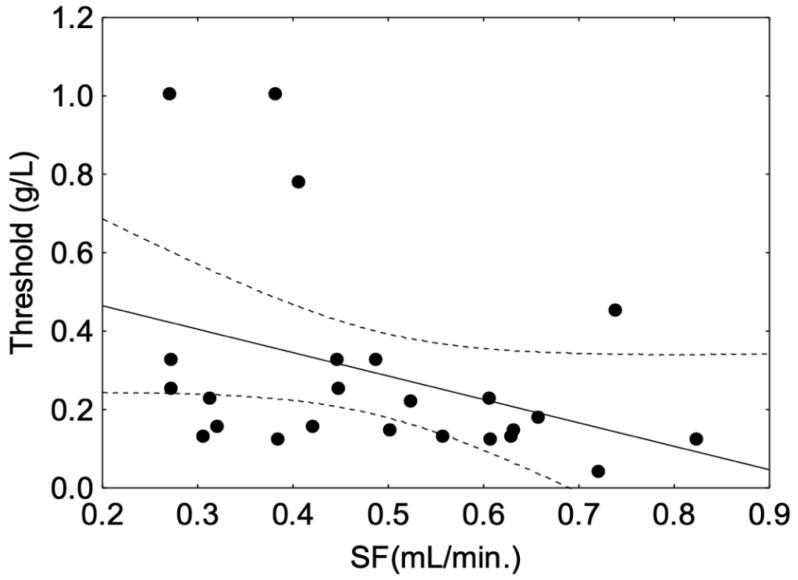
Spearman correlation between astringency threshold and whole salivary flow (SF) observed in the group of young panellists. The plain line corresponds to fitted data. The dotted line corresponds to the confidence interval at 95%. The black dot symbols correspond to individual data points.

**Table 1 molecules-27-01617-t001:** Characteristics of the young (Y) and elderly (O) panels.

Y (n = 24, 18 Males/6 Females)					O (n = 30, 16 Males/14 Females)			
Characteristics	Mean	Median	Range	SD	Mean	Median	Range	SD
Age (years)	29.4	30	24–35	3.8	75	73.5	70–87	4.23
SF (mL/min)	0.49	0.47	0.27–0.82	0.16	0.42	0.35	0.11–0.92	0.23
Protein amount (mg/mL)	0.6	0.62	0.17–1.4	0.27	0.62	0.6	0.22–1.35	0.24
Threshold (g/L)	0.29	0.2	0.04–1.00	0.26	0.41	0.35	0.06–0.78	0.24

SD: standard deviation of the mean; SF: salivary flow.

**Table 2 molecules-27-01617-t002:** Spearman correlation coefficient (r) and *p* value of the astringency threshold and salivary characteristics for the whole (W), young (Y) and elderly (O) panellists. SF: salivary flow.

	SF			Protein Amount		
	W	Y	O	W	Y	O
**Threshold**	r = −0.16 *p* = 0.24	r = 0.44 *p* = 0.03	r = 0.14 *p* = 0.47	r = 0.19 *p* = 0.16	r = 0.18 *p* = 0.39	r = 0.19 *p* = 0.30

## Data Availability

The original data presented in this study are available upon request. However, because of ethical or privacy issues, they will be provided in aggregated form.

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
