# Peer review of "Astringency Sensitivity to Tannic Acid: Effect of Ageing and Saliva"

_molecules, 2022, doi:10.3390/molecules27051617_

Round 1

Reviewer 1 Report

This paper is a very new and innovative one. 

Molecules magazine published a large number of chemical articles, this article is based on the population sensory experiment, although the chemical contents is not much, but to guide the research of flavor chemistry were significant, and many scientists who study the astringency can reference this on astringency related mechanism, especially the saliva protein, saliva flow.

I'd love to see this article published on the cover.

Author Response

We would like to answer the reviewer for his/her very positive comment.

Reviewer 2 Report

Conceptually simple yet experimentally incisive, this manuscript provides definitive evidence regarding astringency perception as a function of age. It also opens new avenues for future reseaches aiming to explore, at the molecular and cellular level, the relationship between salivary compositional/microbiological profile and astrigency sensitivity during aging. Well written and organized, this manuscript definitely deserves to be published. Some minor comments: (1) there is a missing dot on line 46; (2) despite intuitive, it would be easier if readers were introduced to the "O" and "Y" abbreviations right on line 91.

Author Response

We would like to thank the reviewer for his/her very positive comments.

We made the suggested minor modifications accordingly. 

Reviewer 3 Report

The Authors investigate the changes in the astringency perception as a function of age. This is a topical issue, since the number of elderly citizens is rising in many regions of the world. Food product developers are facing challenges when an approximate level of astringency have to be set up for a given product. This study might contribute to the better knowledge in this field.

When reading the paper, I was missing the ‘Materials and methods’ section. Later on, at chapter 5. I found it. Please clarify it with the Journal’s guide, which sequence of chapters is accepted.

Line 91: Group Y and Group O probably refers to the young and older participants, please clarify these labels at their first use.

At Figure 1. the outliers are mentioned. How do Authors see the possibility of dealing with outlier data in that study? Should outliers be removed as ‘dataset cleaning’, or those data points provide richness of the original data matrix and have to be kept as they are?

At Figure 2. the distribution of the salivary flow at the elderly group look like a possible bi-modal distribution. Is this a normal phenomenon in this type of research, or it is a new scientific finding?

Otherwise the study is accurately designed, gives plenty of reflection to the previous results of other researchers, even if those are sometimes contradicting each other.

Author Response

We would like to thank the reviewer for his/her comment and helpful suggestions. 

  • When reading the paper, I was missing the ‘Materials and methods’ section. Later on, at chapter 5. I found it. Please clarify it with the Journal’s guide, which sequence of chapters is accepted. 

We  followed the template provided by the editor. This template recommend a Materials and Methods section at the end of the paper.

  • Line 91: Group Y and Group O probably refers to the young and older participants, please clarify these labels at their first use.                             

The correction has been done

  • At Figure 1. the outliers are mentioned. How do Authors see the possibility of dealing with outlier data in that study? Should outliers be removed as ‘dataset cleaning’, or those data points provide richness of the original data matrix and have to be kept as they are?

The presence of outliers is important to highlight the variability that can be observed in the population. Thus, we preferred to keep them in the dataset. We deal with outliers by performing nonparametric statistics that are more robust to their presence. We gave precisions on this point in the statistic part in the Material and Method section (line 394-400) with appropriate references. 

  • At Figure 2. the distribution of the salivary flow at the elderly group look like a possible bi-modal distribution. Is this a normal phenomenon in this type of research, or it is a new scientific finding?

Variability and heterogeneity in salivary flow is commonly observed in the elderly population. This variability can be due to life-style factors such as smoking habit for instance but also specific aging factors such as hydration status or structural changes in the salivary glands. We add in the result section a little paragraph discussing this point (lines 124-128) with an appropriate reference.

Reviewer 4 Report

Dear Authors,

I have reviewed the manuscript from Wang M. et al., titled "Astringency sensitivity to tannic acid: effect of ageing and saliva". I found it interesting, especially concerning the aim and methodology applied. I think this work is valuable and should be granted acceptance for publication after some minor revisions are made.

My only concern is the presentation of the results in Figure 4. Although a significant Spearman correlation model was found, I am concerned that this might be just the result of the strong influence of a non-negligible number of outlier samples, well-observable in the plot well outside the CI. May I kindly suggest to apply a robust correlation model, so to counteract the potential strong effects of those outliers? Accordingly, this could be commented in the related paragraph, in both cases that a significant robust correlation was found or not. In the latter case, any info on other possibly known factors that might have produced those outliers could be also interesting.

Kind Regards

Author Response

I have reviewed the manuscript from Wang M. et al., titled "Astringency sensitivity to tannic acid: effect of ageing and saliva". I found it interesting, especially concerning the aim and methodology applied. I think this work is valuable and should be granted acceptance for publication after some minor revisions are made.

We would like to thank the reviewer for his/her comment and helpful suggestions. 

My only concern is the presentation of the results in Figure 4. Although a significant Spearman correlation model was found, I am concerned that this might be just the result of the strong influence of a non-negligible number of outlier samples, well-observable in the plot well outside the CI. May I kindly suggest to apply a robust correlation model, so to counteract the potential strong effects of those outliers? Accordingly, this could be commented in the related paragraph, in both cases that a significant robust correlation was found or not. In the latter case, any info on other possibly known factors that might have produced those outliers could be also interesting.

At first, the presence of outliers is important to highlight the variability that can be observed in the population. Thus, we preferred to keep them in the dataset. We deal with outliers by performing nonparametric statistics that are more robust to their presence. Regarding figure 4, Spearman correlation is more appropriate and more robust to the presence of outliers compared to parametric model such as Pearson correlation for instance because it is principally based on the rank  (Croux & Dehon, 2010; De Winter, Gosling, & Potter, 2016; Myers & Sirois, 2004).

We gave precisions on this point in the statistic part in the Material and Method section (line 394-400).

Moreover, the variability and the presence of outliers can be due to life-style factors such as smoking habit for instance but also specific aging factors such as hydration status or structural changes in the salivary glands. We add in the Results section a little paragraph discussing this point (lines 124-128).

At last, we highlight in the Limitation section a little comment regarding the potential effect of these outliers on the statistical results (lines 261-265). 

References:

Croux, C., & Dehon, C. (2010). Influence functions of the Spearman and Kendall correlation measures. Statistical methods & applications, 19(4), 497-515.

De Winter, J. C., Gosling, S. D., & Potter, J. (2016). Comparing the Pearson and Spearman correlation coefficients across distributions and sample sizes: A tutorial using simulations and empirical data. Psychological Methods, 21(3), 273.

Myers, L., & Sirois, M. J. (2004). Spearman correlation coefficients, differences between. Encyclopedia of statistical sciences, 12.